# Constrained Generation of Semantically Valid Graphs via Regularizing Variational Autoencoders

**Tengfei Ma**[*]    **Jie Chen**[*]    **Cao Xiao**
IBM Research
Tengfei.Ma1@ibm.com,    {chenjie,cxiao}@us.ibm.com

## Abstract

Deep generative models have achieved remarkable success in various data domains, including images, time series, and natural languages. There remain, however, substantial challenges for combinatorial structures, including graphs. One of the key challenges lies in the difficulty of ensuring semantic validity in context. For example, in molecular graphs, the number of bonding-electron pairs must not exceed the valence of an atom; whereas in protein interaction networks, two proteins may be connected only when they belong to the same or correlated gene ontology terms. These constraints are not easy to be incorporated into a generative model. In this work, we propose a regularization framework for variational autoencoders as a step toward semantic validity. We focus on the matrix representation of graphs and formulate penalty terms that regularize the output distribution of the decoder to encourage the satisfaction of validity constraints. Experimental results confirm a much higher likelihood of sampling valid graphs in our approach, compared with others reported in the literature.

## 1   Introduction

The recent years have witnessed rapid progress in the development of deep generative models for a wide variety of data types, including continuous data (e.g., images [39]) and sequences (e.g., time series [38] and natural language sentences [18]). Representative methods, including generative adversarial networks (GAN) [16] and variational autoencoders (VAE) [25], learn a distribution parameterized by deep neural networks from a set of training examples. Amid the tremendous progress, deep generative models for combinatorial structures, particularly graphs, are less mature. The difficulty, in part, is owing to the challenge of an efficient parameterization of the graphs while maintaining semantic validity in context. Whereas there exists a large body of work on learning an abstract representation of a graph [34, 11, 26, 10, 14], how one decodes it into a valid native representation is less straightforward.

One natural line of approaches [24, 29] treats the generation process as sequential decision making, wherein nodes and edges are inserted one by one, conditioned on the graph constructed so far. Vectorial representations of the graph, nodes, and edges may be simultaneously learned. When the training graphs are large, however, learning a long sequence of decisions is reportedly challenging [29]. Moreover, existing work is limited to a predefined ordering of the sequence, leaving open the role of permutation.

Another approach [37, 36] is to build a probabilistic graph model based on the matrix representation. In the simplest form, the adjacency matrix may encode the probability of the existence of each edge, with additionally a probability vector indicating the existence of nodes. The challenge of such an approach, in contrast to the local decisions made in the preceding one, is that global properties (e.g.,

---

[*]Equal contribution.

connectivity) of the graph are hard to control [36]. Furthermore, in applications, there often exist constraints demanding that only certain combinations of the nodes and edges are valid. For example, in molecular graphs, the number of bonding-electron pairs cannot exceed the valence of an atom. Translated to the graph language, this constraint means that for a node of certain type, the incident edges collectively must have total type values not exceeding a threshold, where the "type value" is some numeric property of an edge type. For another example, in protein interaction networks, two proteins may be connected only when they belong to the same or correlated gene ontology terms. That is, two nodes are connected only when their types are compatible. These constraints are difficult to satisfy when the sampling of nodes and edges is independent according to the probability matrix. Such challenges motivate the present work.

In this work, we propose a regularization framework for VAEs to generate semantically valid graphs. We focus on the matrix representation and formulate penalty terms that address validity constraints. The penalties in effect regularize the distributions of the existence and types of the nodes and edges collectively. Examples of the constraints include graph connectivity, node label compatibility, as well as valence in the context of molecular graphs. All these constraints are formulated with respect to the node-label matrix and the edge-label tensor. We demonstrate the high probability of generating valid graphs under the proposed framework with two benchmark molecule data sets and a synthetic node-compatible data set.

## 2    Related Work

Generative models aim at learning the distribution of training examples. The emergence of deep architectures for generative modeling, including VAE [25], GAN [16], and Generative Moment Matching Networks (GMMN) [28], demonstrates state-of-the-art performance in various data domains, including continuous ones (e.g., time series [38], musical improvisation [22], image generation [31], and video synthesis [40]) and discrete ones (e.g, program induction [13] and molecule generation [15, 27]). They inspire a rich literature on the training and extensions of these models; see, e.g., the work [8, 1, 3, 30, 33, 7, 19, 2, 17].

Prior to deep learning, there exist several random graph models, notably the Erdős–Rényi model [12] and Barabasi's scale-free model [4], that explicitly describe a class of graphs. These models are limited to specific graph properties (e.g., degree distribution) and are not sufficiently expressive for real-life graphs with richer structures.

Recently, deep generative models for graphs are attracting surging interests. In GraphVAE [36], a work the most relevant to ours, the authors use VAE to reconstruct the matrix representation of a graph. Based on this representation, our work formulates a regularization framework to impose constraints so that the sampled graphs are semantically valid. In another work, NetGAN [6], the authors use the Wasserstein GAN [2] formulation to generate random paths and assemble them for a new graph. Rather than learning a distribution of graphs, this work learns the connectivity structure of a single graph and produces a sibling graph that shares the distribution of the paths. Yet another approach for graph generation is to produce nodes and edges sequentially [24, 29]. This approach resembles the usual sequence models for texts, wherein new tokens are generated conditioned on the history. These models are increasingly difficult to train as the sequence becomes longer and longer.

Application-wise, *de novo* design of molecules is a convenient testbed and an easily validated task for deep graph generative models. Molecules can be represented as undirected graphs with atoms as nodes and bonds as edges. Most of the work for molecule generation, however, is not applicable to a general graph. For example, a popular approach to tackling the problem is based on the SMILES [41] representation. An array of work designs recurrent neural net architectures for generating SMILES strings [15, 5, 35]. Gómez-Bombarelli et al. [15] use VAEs to encode SMILES strings into a continuous latent space, from which one may search for new molecules with desirable properties through Bayesian optimization. The string representation, however, is very brittle, because small changes to the string may completely violate chemical validity. To resolve the problem, Kusner et al. [27] propose to use the SMILES grammar to generate a parse tree, which in turn is flattened as a SMILES string. This approach guarantees the syntactic validity of the output, but semantic validity is still questionable. Dai et al. [9] further apply attribute grammar as a constraint in the parse-tree generation, a step toward semantic validity. Jin et al. [23] exploit the fact that molecular graphs

may be turned into a tree by treating the rings as super nodes. We note that none of these methods generalizes to a general graph.

# 3 Regularized Variational Autoencoder for Graphs

In this section, we propose a regularization framework for VAEs to generate semantically valid graphs. The framework is inspired by the transformation of a constrained optimization problem to a regularized unconstrained one.

## 3.1 Graph Representation and Probability Model

Let a collection of graphs have at most $N$ nodes, $d$ node types, and $t$ edge types. A graph from this collection is normally represented by the tuple $(F, E)$ with

$$\text{node-label matrix } F \in \mathbb{R}^{N \times (1+d)} \quad \text{and} \quad \text{edge-label tensor } E \in \mathbb{R}^{N \times N \times (1+t)},$$

where 0-based indexing is used for convenience. The node types range from 1 to $d$ and the edge types from 1 to $t$. For each node $i$, the row $F(i, :)$ is one-hot. If $F(i, 0) = 1$, the node is nonexistent (subsequently called "ghost nodes"). Otherwise, the "on" location of $F(i, :)$ indicates the label of the node. Similarly, for each pair $(i, j)$, the fiber $E(i, j, :)$ is one-hot. If $E(i, j, 0) = 1$, the edge is nonexistent; otherwise, the "on" location of the fiber indicates the label of the edge.

We relax the one-hot rows of $F$ and fibers of $E$ to probability vectors and write with a tilde notation $\widetilde{G} = (\widetilde{F}, \widetilde{E})$. Now, $\widetilde{F}(i, r)$ is the probability of node $i$ belonging to type $r$ (nonexistent if $r = 0$), and $\widetilde{E}(i, j, k)$ is the probability of edge $(i, j)$ belonging to type $k$ (nonexistent if $k = 0$). Then, $\widetilde{G}$ is a random graph model, from which one may generate random realizations of graphs. Under the independence assumption, the probability of sampling a graph $G$ using the model $\widetilde{G}$ is

$$\prod_{i=1}^{N} \prod_{r=1}^{1+d} \widetilde{F}(i, r)^{F(i,r)} \prod_{i<j} \prod_{k=1}^{1+t} \widetilde{E}(i, j, k)^{E(i,j,k)}. \tag{1}$$

In what follows, we use the one-hot $G$ to denote a graph and the probabilistic $\widetilde{G}$ to denote the sampling distribution.

## 3.2 Variational Autoencoder

The goal of a generative model is to learn a probability distribution from a set of training graphs, such that one can sample new graphs from it. To this end, let $z$ be a latent vector. We want to learn a latent model $p_\theta(G|z)$, which is defined by a generative network with parameters $\theta$ and output $\widetilde{G}$. Assuming independence of training examples, the objective, then, is to maximize the log-evidence of the data:

$$\sum_l \log p_\theta(G^{(l)}) = \sum_l \log \int p_\theta(G^{(l)}|z) p_\theta(z) \, dz, \tag{2}$$

where the superscript $l$ indexes training examples.

The integral (2) being generally intractable, a common remedy in variational Bayes is to use a variational posterior $q_\phi(z|G)$, defined by an inference network with parameters $\phi$, to approximate the actual posterior $p_\theta(z|G)$. In this vein, the log-evidence (2) admits a lower bound

$$L_{\text{ELBO}} = -\sum_l D_{\text{KL}}\Big(q_\phi(z|G^{(l)}) \,\|\, p_\theta(z)\Big) + \sum_l \mathbb{E}_{q_\phi(z|G^{(l)})}\Big[\log p_\theta(G^{(l)}|z)\Big], \tag{3}$$

where $D_{\text{KL}}$ denotes the Kullback–Leibler divergence. We maximize the lower bound $L_{\text{ELBO}}$ with respect to $\theta$ and $\phi$.

Such a variational treatment lands itself to an autoencoder, where the inference network encodes a training example $G$ into a latent representation $z$, and the generative network decodes the latent $z$ and reconstructs a graph from the probabilistic model $\widetilde{G}$, such that it is as close to $G$ as possible. Between the two constituent parts of $L_{\text{ELBO}}$ in (3), the expectation term is the negative reconstruction loss,

whereas the KL divergence term serves as a regularization that encourages the variational posterior to stay close with the prior.

It remains to define the probability distributions. The likelihood $p_\theta(G|z)$ simply follows the probability model (1). The variational posterior $q_\phi(z|G)$ is a factored Gaussian with mean vector $\mu$ and variance vector $\sigma^2$. Usually, the prior $p_\theta(z)$ is standard normal, but we find that parameterizing it with a trainable mean vector $m$ and variance vector $s^2$ sometimes improves inference.

### 3.3 Regularization

The central contribution of this work is an approach to imposing validity constraints in the training of VAEs. In optimization, (in)equality constraints may be moved to the objective function to form a Lagrangian function, whose solution coincides with that of the original objective. This connection is one of the justifications of using regularization to formulate an unconstrained objective, which is otherwise challenging to optimize. The regularization corresponds to the original (in)equality constraints. For example, it is well known that the Euclidean-ball constraint is, under certain conditions, equivalent to an $L_2$ regularization.

For VAE, we want the samples produced by the generative network $p_\theta(G|z)$ to be valid, regardless of what latent value $z$ one starts with. The constraint set is then infinite because of the cardinality of the (often) continuous random variable $z$. Hence, we first generalize the Lagrangian function for an infinite constraint set, and then use the generalization to motivate a sound approach for formulating regularization. The idea turns out to be fairly simple—it suffices to marginalize the constraints over $z$.

Let $f(x)$ be the objective function to be minimized. For VAE, the unknown $x$ includes both the generative parameter $\theta$ and the variational parameter $\phi$, and $f$ is the negative lower bound $-L_{\text{ELBO}}$. Suppose that for each $z$ there are $m$ equality constraints and $r$ inequality constraints, such that the problem is formally written as

$$
\begin{aligned}
\min_x \quad & f(x) \\
\text{subject to} \quad & \text{for almost all } z \sim p_x(z), \\
& h_1(x, z) = 0, \ldots, h_m(x, z) = 0, \\
& g_1(x, z) \leq 0, \ldots, g_r(x, z) \leq 0.
\end{aligned}
\tag{4}
$$

The phrase "almost all" means that the set of $z$ violating the constraints has a zero measure.

To solve (4), we generalize the usual notion of Lagrangian function to

$$
\mathcal{L}(x, \lambda, \mu) = f(x) + \sum_{i=1}^{m} \lambda_i \widetilde{h}_i(x) + \sum_{j=1}^{r} \mu_j \widetilde{g}_j(x),
\tag{5}
$$

where $\{\lambda_i\}$ and $\{\mu_j\}$ are Lagrangian multipliers and

$$
\widetilde{h}_i(x) = \left[ \int h_i(x, z)^2 p_x(z) \, dz \right]^{\frac{1}{2}} \quad \text{and} \quad \widetilde{g}_j(x) = \left[ \int g_j(x, z)^2 p_x(z) \, dz \right]^{\frac{1}{2}}.
\tag{6}
$$

These two tilde terms correspond to the marginalization of the squared constraints; hence, the dependency on $z$ in the Lagrangian function is eliminated. After a technical definition, we give a theorem that resembles the usual KKT condition for constrained problems. Its proof is given in the supplementary material.

**Definition 1.** For any feasible $x$, denote by $A(x) = \{j \mid g_j(x, z) = 0 \text{ for almost all } z\} = \{j \mid \widetilde{g}_j(x) = 0\}$ the set of active inequality constraints. A feasible $x$ is said to be *regular* if the equality constraint gradients $\nabla \widetilde{h}_i(x)$, $i = 1, \ldots, m$, and the active inequality constraint gradients $\nabla \widetilde{g}_j(x)$, $j = 1, \ldots, r$, are linearly independent.

**Theorem 1.** *Let $x^*$ be a local minimum of the problem* (4) *and assume that $x^*$ is regular. Then, there exist unique vectors $\lambda^* = (\lambda_1^*, \ldots, \lambda_m^*)$ and $\mu^* = (\mu_1^*, \ldots, \mu_r^*)$ such that (a) $\nabla_x \mathcal{L}(x^*, \lambda^*, \mu^*) = 0$; (b) $\mu_j^* \geq 0$, $j = 1, \ldots, r$; and (c) $\mu_j^* = 0$, $\forall j \notin A(x^*)$.*

The above theorem indicates that a solution $x^*$ of the constrained problem (4) coincides with that of the unconstrained minimization of (5). An intuitive explanation of why validity constraints for

every $z$ may be equivalently reformulated as regularization terms involving only the marginalization of $z$, is that $h_i(x, z)$ is zero for almost all $z$ if and only if $\widetilde{h}_i(x)$ is zero. Moreover, active inequality constraints are equivalent to equality ones, and nonactive constraints have multipliers equal to zero. This argument proves a majority of the conclusions in Theorem 1. The spirit is that marginalization is a powerful tool for formulating regularizations that faithfully represent the constraints.

Note that the squaring of $h_i$ (and similarly of $g_j$) in (6) ensures that $h_i(x, z)$ is zero for almost all $z$ if and only if $\widetilde{h}_i(x)$ is zero, a premise of the correctness of the theorem. Without squaring, this if-and-only-if statement does not hold.

### 3.4 Training

Returning to the notation of VAE, let us write the $i$-th validity constraint as $g_i(\theta, z) \leq 0$ for all $z$. Based on the preceding subsection, the loss function for training VAE may then be written as

$$-L_{\text{ELBO}}(\theta, \phi) + \mu \sum_i \left[ \int g_i(\theta, z)^2 p_\theta(z) \, dz \right]^{\frac{1}{2}},$$

where $\mu \geq 0$ is treated as a tunable hyperparameter and the square-bracket term is a regularization. We avoid using different $\mu$'s for each $i$ to reduce the number of hyperparameters. A problem for this loss function is that it penalizes not only the undesirable case $g_i(\theta, z) > 0$, but also the opposite desirable case $g_i(\theta, z) < 0$, because of the presence of the square. Hence, we make a slight modification to the regularization and use the following loss function for training instead:

$$-L_{\text{ELBO}}(\theta, \phi) + \mu \sum_i \left[ \int g_i(\theta, z)_+^2 p_\theta(z) \, dz \right]^{\frac{1}{2}}, \tag{7}$$

where $g_+ = \max(g, 0)$ denotes the ramp function. This regularization will not penalize the desirable case $g_i(\theta, z) \leq 0$.

Ad hoc as it may sound, the use of the ramp function follows the same interpretation of the usual relationship between a constrained optimization problem and the corresponding regularized unconstrained one. In the usual KKT condition where an inequality constraint $g \leq 0$ is not squared, the nonnegative multiplier $\mu$ ensures that the regularization $\mu g$ penalizes the undesirable case $g > 0$. Here, on the other hand, the squaring of the inequality constraint cannot distinguish the sign of $g$ anymore. Therefore, $g_+$ is a correct replacement.

In practice, the integral in the regularization may be intractable, and hence we appeal to Monte Carlo approximation for evaluating the loss in each parameter update:

$$-L_{\text{ELBO}}(\theta, \phi) + \mu \sum_i g_i(\theta, z)_+, \quad \text{where} \quad z \sim p_\theta(z). \tag{8}$$

Such an approach is similar to the training of standard VAE, where the expectation term in (3) is also approximated by a Monte Carlo sample. There are two important distinctions, however. First, in the standard VAE, the latent vector $z$ is sampled from the variational posterior $q_\phi(z|G)$, whereas in regularized VAE, the additional $z$ is sampled from the prior $p_\theta(z)$. Second, the variational posterior sample $z$ is decoded from a training graph $G$, whereas the prior sample $z$ does not come from any training graph. We call the latter $z$ *synthetic*. Despite the distinctions, reparameterization must be used for sampling in both cases, so that $z$ is differentiable.

This training procedure is schematically illustrated by Figure 1. We use $l$ to index a training example and $\underline{l}$ (note the underline) to denote a synthetic example, needed by regularization. The top flow denotes the standard VAE, where an input graph $G^{(l)}$ is encoded as $z^{(l)}$, which in turn is decoded to compute the ELBO loss. The bottom flow denotes the regularization, where a synthetic $z^{(\underline{l})}$ is decoded to compute the constraints $g_i(\theta, z^{(\underline{l})})_+$. The combination of the two gives the total loss in each optimization step.

## 4 Constraint Formulation

In this section we formulate several constraints $g_i(\theta, z)$ used as regularization in (7). All these constraints are concerned with the decoder output $\widetilde{G} = (\widetilde{F}, \widetilde{E})$ and hence the dependencies on $\theta$ and

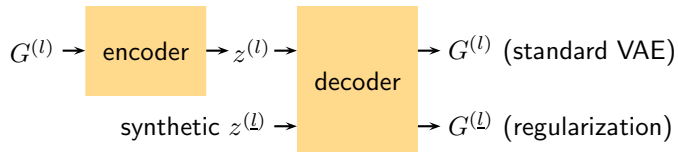

Figure 1: Overview of the regularization framework. In addition to a standard VAE (top flow), regularizations are imposed on synthetic $z^{(l)}$ sampled from the prior (bottom flow).

$z$ are omitted for readability; that is, we write the constraints as $g_i$. In this case, $i$ corresponds to a graph node. If the constraints are imposed on each edge $(i, j)$, we write the constraints as $g_{ij}$.

### 4.1 Ghost Nodes and Valence

A ghost node has no incident edges. On the other hand, if a graph represents a molecule, the configuration of the bonds must meet the valence criteria of the atoms. These two seemingly unrelated constraints have a common characteristic: the edges incident to a node are collectively subject to a limited choice of existence and types.

Denote by $V(i)$ the capacity of a node $i$ and by $U(i)$ an upper bound of the capacity. For example, $V$ is the number of bonding-electron pairs and $U$ is the valence. The constraint is written as

$$g_i = V(i) - U(i). \tag{9}$$

To define $V$ and $U$, let $h(k)$ be the capacity function of an edge type $k$:

$$h(\text{nonexistent}) = 0, \quad h(\text{single bond}) = 1, \quad h(\text{double bond}) = 2, \quad h(\text{triple bond}) = 3.$$

Then,

$$V(i) = \sum_{j \neq i} \sum_k h(k) \widetilde{E}(i, j, k).$$

This expression reads that if the fiber $\widetilde{E}(i, j, :)$ is one hot, the inner summation is exactly the type capacity of the edge $(i, j)$, and the outer summation sums over all other nodes $j$ in the graph, forming the overall capacity of the node $i$. Of course, $\widetilde{E}(i, j, :)$ is not one-hot; hence, the inner summation is the expected capacity for the edge. A similar expectation is used to define the upper bound $U$ (valence):

$$U(i) = \sum_r u(r) \widetilde{F}(i, r), \quad \text{where} \quad u(r) = \begin{cases} \text{valence of node type } r, & \text{if } r \neq 0, \\ 0, & \text{if } r = 0. \end{cases}$$

Note that for graphs other than molecules, where the concept of valence does not exist, ghost nodes still must obey the constraint (9). In such a case, the capacity function $h$ is 0 if an edge is nonexistent and 1 otherwise. The function $u$ is 0 when $r = 0$ and $N - 1$ when $r \neq 0$.

### 4.2 Connectivity

A graph is connected if there is a path between every pair of non-ghost nodes. If $A$ is the adjacency matrix of the graph, then the $(i, j)$ element of the matrix $B = I + A + A^2 + \cdots + A^{N-1}$ is nonzero if and only if $i$ and $j$ are connected by a path. Let $q$ be an indicator vector of non-ghost nodes; that is, $q(i) = 0$ if $i$ is a ghost node and $= 1$ otherwise. Then, the graph must satisfy the following constraint:

$$q(i)q(j) \cdot \mathbf{1}\{B(i, j) = 0\} + [1 - q(i)q(j)] \cdot \mathbf{1}\{B(i, j) \neq 0\} \leq 0, \quad \forall i \neq j.$$

In words, if at least one of $i$ and $j$ is a ghost node, $B(i, j)$ must be zero. On the other hand, if neither of $i$ and $j$ is a ghost node, $B(i, j)$ must be nonzero.

The above constraint is unfortunately nondifferentiable. To formulate a differentiable version, we first let $q(i) = 1 - \widetilde{F}(i, 0)$, because $\widetilde{F}(i, 0)$ is the probability that node $i$ is a ghost node. Then, for the matrix $B$, we need to define $A$. Because $\widetilde{E}(i, j, 0)$ is the probability that edge $(i, j)$ is nonexistent, we let $A(i, j) = 1 - \widetilde{E}(i, j, 0)$. Because now $A$ is probabilistic, an element $(i, j)$ of $A$ is nonzero even

if the probability that $i$ and $j$ are connected is tiny. Such a tiny nonzero element will be amplified through taking powers of $A$. Hence, we need a sigmoid transform for each power of $A$ to suppress amplification. Specifically, define $\sigma(x) = \left\{1 + \exp[-a(x - \frac{1}{2}))]\right\}^{-1}$, where $a > 0$ is sufficiently large to make most of the transformed mass close to either 0 or 1 (say, $a = 100$). Then, define

$$A_0 = I, \quad A_1 = A, \quad A_{i+1} = \sigma(A_i A), \ i = 1, \ldots, N-2, \quad B = \sum_{i=0}^{N-1} A_i, \quad C = \sigma(B),$$

where $C(i,j)$ is now sufficiently close to the indicator $\mathbf{1}\{B(i,j) \neq 0\}$. Thus, the differentiable version of the constraint is

$$g_{ij} = q(i)q(j) \cdot [1 - C(i,j)] + [1 - q(i)q(j)] \cdot C(i,j). \tag{10}$$

### 4.3 Node Compatibility

A compatibility matrix $D \in \mathbb{R}^{(1+d) \times (1+d)}$ summarizes the compatibility of every pair of node types. When both types $r$ and $r'$ are nonzero, $D(r, r')$ is 1 if the two types are compatible and 0 otherwise. Moreover, ghost nodes are incompatible. Hence, when either of $r$ and $r'$ is zero, $D(r, r') = 0$.

We now consider a constraint that mandates that two nodes are connected only when their node types are compatible. This constraint appears in, for example, protein interaction networks where two proteins are connected only when they belong to the same or correlated gene ontology terms. Consider the matrix $P = \widetilde{F}D\widetilde{F}^T$. The $(i,j)$ element of $P$ is the probability that nodes $i$ and $j$ have compatible types. We want node pairs with low compatibility to be disconnected. Hence, the constraint is

$$g_{ij} = [1 - \widetilde{E}(i, j, 0)][1 - P(i, j)] - \alpha, \tag{11}$$

where $\alpha \in (0, 1)$ is a tunable hyperparameter. The interpretation of (11) is as follows. In order to satisfy the constraint $g_{ij} \leq 0$, the probability that edge $(i,j)$ exists, $1 - \widetilde{E}(i, j, 0)$, must be $\leq \alpha/[1 - P(i,j)]$. The smaller is $P(i,j)$, the lower is the threshold, which leads to a smaller probability for the edge to exist. On the other hand, once $P(i,j)$ exceeds $1 - \alpha$, we see that $g_{ij} \leq 0$ always holds regardless of the existence probability $1 - \widetilde{E}(i, j, 0)$. Hence, for highly compatible pairs, the corresponding edge may or may not exist.

## 5 Experiments

### 5.1 Tasks, Data Sets, and Baselines

We consider two tasks: the generation of molecular graphs and that of node-compatible graphs. For molecular graphs, two benchmark data sets are QM9 [32] and ZINC [21]. The former contains molecules with at most 9 heavy atoms whereas the latter consists of drug-like commercially available molecules extracted at random from the ZINC database.

For node-compatible graphs, we construct a synthetic data set by first generating random node labels, followed by connecting node pairs under certain probability if their labels are compatible. The compatibility matrix $D$, ignoring the empty top row and left column, is

$$\begin{bmatrix} 0 & 1 & 1 & 1 & 0 \\ 1 & 0 & 1 & 0 & 1 \\ 1 & 1 & 0 & 1 & 1 \\ 1 & 0 & 1 & 0 & 0 \\ 0 & 1 & 1 & 0 & 0 \end{bmatrix}.$$

Based on this matrix, we generate 100,000 random node-compatible graphs. For each graph, the number of nodes is a uniformly random integer $\in [10, 15]$. Each node is randomly assigned one of the five possible labels. Then, for each pair of nodes whose types are compatible according to $D$, we assign an edge with probability 0.4. The graph is not necessarily connected.

Table 1 summarizes the data sets.

Baselines for molecular graphs are character VAE (CVAE) [15] and grammar VAE (GVAE) [27]. Both methods are based on the SMILES string representation. The codes are downloaded from

Table 1: Data sets.

| | # Graphs | # Nodes | # Node Types | # Edge Types |
|---|---|---|---|---|
| QM9 (molecule) | 134k | 9 | 4 | 3 |
| ZINC (molecule) | 250k | 38 | 9 | 3 |
| Node-compatible | 100k | 15 | 5 | 1 |

`https://github.com/mkusner/grammarVAE`. For node-compatible graphs, there is no baseline, because it is a new task. However, we will compare the results of using regularization versus not and show the effectiveness of the proposed method.

## 5.2 Network Architecture

For input representation, we unfold the edge-label tensor $E \in \mathbb{R}^{N \times N \times (1+t)}$ and concatenate it with the node-label matrix $F \in \mathbb{R}^{N \times (1+d)}$ to form a wide matrix with $N$ rows and $(1+d) + N(1+t)$ columns. The encoder is a 4-layer convolutional neural net (32, 32, 64, 64 channels with filter size 3×3). The latent vector $z$ is normally distributed and its mean and variance are generated from two separate fully connected layers over the output of the CNN encoder. The decoder follows the generator of DCGAN [31] and is a 4-layer deconvolutional neural net (64, 32, 32, 1 channels with filter size 3×3). Both the encoder and the decoder have modules of the form Convolution–BatchNorm–ReLU [20].

## 5.3 Results

**Effect of Regularization.** Table 2 compares the performance of standard VAE with that of the proposed regularized VAE. The column "% Valid" denotes the percentage of valid graphs among those sampled from the prior, and the column "ELBO" is the lower bound approximation of the log-evidence of the training data. Regularization parameters are tuned for the highest validity. One sees that regularization noticeably boosts the the validity percentage in all data sets. ELBO becomes lower, as expected, because optimal parameters for the standard objective are not optimal for the regularized one. However, the difference of the two ELBOs is not large.

Table 2: Standard VAE versus regularized VAE.

| QM9 | | | ZINC | | | Node-compatible | | |
|---|---|---|---|---|---|---|---|---|
| Method | % Valid | ELBO | Method | % Valid | ELBO | Method | % Valid | ELBO |
| Standard | 83.2 | -17.3 | Standard | 29.6 | -46.5 | Standard | 40.2 | -42.5 |
| Regul. | 96.6 | -18.5 | Regul. | 34.9 | -47.0 | Regul. | 98.4 | -51.2 |

**Comparison with Baselines.** Validity percentage is not the only metric for measuring the success of an approach. In Table 3 we include other common metrics used in the literature and compare our results with those of the baselines. The column "% Novel" is the percentage of valid graphs sampled from the prior and not occurring in the training set, and the column "% Recon." is the percentage of holdout graphs (in the training set) that can be reconstructed by the autoencoder. Our results substantially improve over those of the character VAE and grammar VAE.

Table 3: Comparison with baselines. Baseline results: The "% Valid" and "% Novel" columns of QM9 are copied from Simonovsky and Komodakis [36]. The "% Valid" and "% Recon." columns of ZINC are copied from Kusner et al. [27]. The "% Recon." column of QM9 and "% Novel" column of ZINC are computed by using the downloaded codes mentioned in Section 5.1.

| QM9 | | | | ZINC | | | |
|---|---|---|---|---|---|---|---|
| Method | % Valid | % Novel | % Recon. | Method | % Valid | % Novel | % Recon. |
| Proposed | 96.6 | 97.5 | 61.8 | Proposed | 34.9 | 100 | 54.7 |
| GVAE | 60.2 | 80.9 | 96.0 | GVAE | 7.2 | 100 | 53.7 |
| CVAE | 10.3 | 90.0 | 3.61 | CVAE | 0.7 | 100 | 44.6 |

**Smoothness of Latent Space.** We visually inspect the coherence of the latent space in two ways. In the first one, randomly pick a graph in the training set and encode it as $z$ in the latent space. Then, decode latent vectors on a grid centering at $z$ and with random orientation. We show the grid of graphs on a two-dimensional plane. In the second approach, randomly pick a few pairs in the training set and for each pair, perform a linear interpolation in the latent space. Figure 2 shows that the transitions of the graphs are quite smooth.

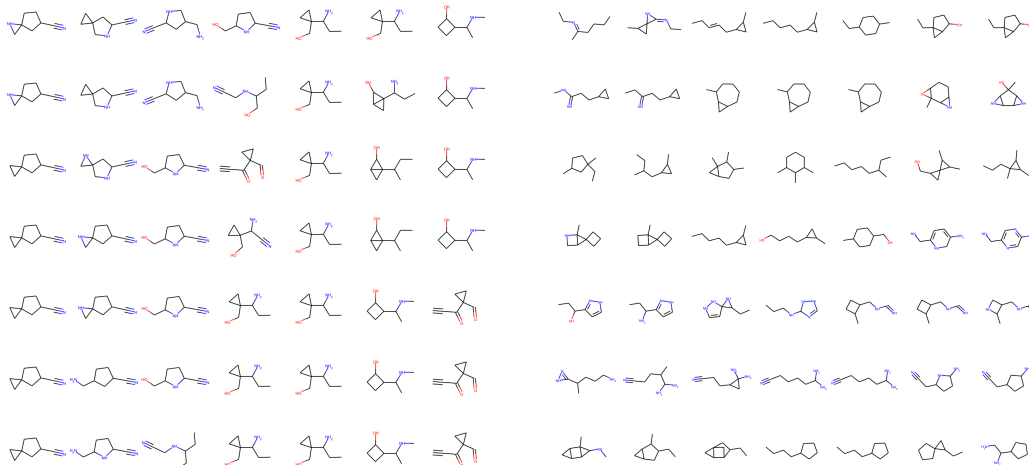

Figure 2: Visualization of latent space. Data set: QM9. Left: Two-dimensional plane. Right: Each row is a one-dimensional interpolation.

**Denoising.** Node-compatible graphs are often noisy. We perform an experiment to show that the proposed regularization may be used to recover a graph that obeys the compatibility constraints. To this end, we generated another set of 10,000 graphs and randomly inserted edges for noncompatible node pairs. Then, we applied regularized VAE to reconstruct the graphs and investigated how many of them were valid. Results are shown in Table 4. One sees that the proposed approach leads to a high probability of reconstructing valid graphs, whereas standard VAE fails in most of the cases.

Table 4: Percentage of validly decoded graphs.

| Standard VAE | Regularized VAE |
|---|---|
| 11.2 | 93.8 |

For more details of all the experiments, the reader is referred to the supplementary material.

## 6   Conclusions

Generating semantically valid graphs is a challenging subject for deep generative models. Whereas substantial breakthrough is seen for molecular graphs, rarely a method is generalizable to a general graph. In this work we propose a regularization framework for training VAEs that encourages the satisfaction of validity constraints. The approach is motivated by the transformation of a constrained optimization problem to a regularized unconstrained one. We demonstrate the effectiveness of the framework in two tasks: the generation of molecular graphs and that of node-compatible graphs.

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
