[Supplementary Material]

# Constrained Generation of Semantically Valid Graphs via Regularizing Variational Autoencoders (Supplementary Material)

**Tengfei Ma**[*]    **Jie Chen**[*]    **Cao Xiao**
IBM Research
Tengfei.Ma1@ibm.com,    {chenjie,cxiao}@us.ibm.com

## 1   Proof of Theorem 1

Let us first label the results of the theorem:

$$\nabla_x \mathcal{L}(x^*, \lambda^*, \mu^*) = 0, \tag{1}$$

and

$$\mu_j^* \geq 0, \quad j = 1, \ldots, r, \tag{2}$$

$$\mu_j^* = 0, \quad \forall j \notin A(x^*). \tag{3}$$

If $x^*$ is a local minimum of (4), it is also a local minimum of the following problem

$$\min_x \quad f(x)$$
$$\text{subject to} \quad \text{for almost all } z \sim p_x(z),$$
$$h_1(x, z) = 0, \ldots, h_m(x, z) = 0,$$
$$g_j(x, z) = 0, \; j \in A(x^*),$$

which is equivalent to

$$\min_x \quad f(x)$$
$$\text{subject to} \quad \widetilde{h}_1(x) = 0, \ldots, \widetilde{h}_m(x) = 0,$$
$$\widetilde{g}_j(x) = 0, \; j \in A(x^*),$$

by the definition of $\widetilde{h}_i$ and $\widetilde{g}_j$. Then, based on a standard result of Lagrange multipliers, there exist $\lambda^* = (\lambda_1^*, \ldots, \lambda_m^*)$ and $\mu_j^*, j \in A(x^*)$, such that

$$\nabla f(x^*) + \sum_{i=1}^{m} \lambda_i^* \nabla \widetilde{h}_i(x^*) + \sum_{j \in A(x^*)} \mu_j^* \nabla \widetilde{g}_j(x^*) = 0.$$

For $j \notin A(x^*)$, adding the gradient terms with respect to $\mu_j^* = 0$ we obtain

$$\nabla f(x^*) + \sum_{i=1}^{m} \lambda_i^* \nabla \widetilde{h}_i(x^*) + \sum_{j=1}^{r} \mu_j^* \nabla \widetilde{g}_j(x^*) = 0, \tag{4}$$

which proves (1) and (3).

---

[*]Equal contribution.

It remains to prove (2). We introduce the functions
$$\widetilde{g}_j^+(x) = \max\{0, \widetilde{g}_j(x)\}, \quad j = 1, \ldots, r,$$
and for each $k = 1, 2, \ldots$, the penalized problem
$$\min_x \quad F^k(x) \equiv f(x) + \frac{k}{2}\|\widetilde{h}(x)\|^2 + \frac{k}{2}\|\widetilde{g}^+(x)\|^2 + \frac{\alpha}{2}\|x - x^*\|^2$$
$$\text{subject to} \quad x \in S,$$
where $\alpha$ is a fixed positive scalar, $S = \{x \mid \|x - x^*\| \le \epsilon\}$, and $\epsilon > 0$ is such that $f(x^*) < f(x)$ for all feasible $x$ with $x \in S$. Note that the function $\widetilde{g}_j^+(x)$ is continuously differentiable with gradient $2\widetilde{g}_j^+(x)\nabla\widetilde{g}_j(x)$. If $x^k$ minimizes $F^k(x)$ over $S$, we have for all $k$,

$$F^k(x^k) = f(x^k) + \frac{k}{2}\|\widetilde{h}(x^k)\|^2 + \frac{k}{2}\|\widetilde{g}^+(x^k)\|^2 + \frac{\alpha}{2}\|x^k - x^*\|^2 \le F^k(x^*) = f(x^*), \quad (5)$$

and since $f(x^k)$ is bounded over $S$, we obtain
$$\lim_{k\to\infty}\|\widetilde{h}(x^k)\| = 0 \quad \text{and} \quad \lim_{k\to\infty}\|\widetilde{g}^+(x^k)\| = 0;$$
otherwise the left-hand side of (5) would become unbounded above as $k \to \infty$. Therefore, every limit point $\overline{x}$ of $\{x^k\}$ satisfies $\widetilde{h}(\overline{x}) = 0$ and $\widetilde{g}^+(\overline{x}) = 0$. Furthermore, (5) yields $f(x^k) + (\alpha/2)\|x^k - x^*\|^2 \le f(x^*)$ for all $k$, so by taking the limit as $k \to \infty$, we obtain
$$f(\overline{x}) + \frac{\alpha}{2}\|\overline{x} - x^*\|^2 \le f(x^*).$$
Since $\overline{x} \in S$ and $\overline{x}$ is feasible, we have $f(x^*) \le f(\overline{x})$, which when combined with the preceding inequality yields $\|\overline{x} - x^*\| = 0$ so that $\overline{x} = x^*$. Thus, the sequence $\{x^k\}$ converges to $x^*$, and it follows that $x^k$ is an interior point of the closed sphere $S$ for sufficiently large $k$. Therefore, $x^k$ is an unconstrained local minimum of $F^k(x)$ for sufficiently large $k$.

For the first order necessary condition, we have for sufficiently large $k$,
$$0 = \nabla F^k(x^k) = \nabla f(x^k) + k\nabla\widetilde{h}(x^k)\widetilde{h}(x^k) + k\nabla\widetilde{g}(x^k)\widetilde{g}^+(x^k) + \alpha(x^k - x^*). \quad (6)$$
Define
$$E(x) = \begin{bmatrix} \nabla\widetilde{h}(x) & \nabla\widetilde{g}(x) \end{bmatrix} \quad \text{and} \quad e(x) = \begin{bmatrix} \widetilde{h}(x) \\ \widetilde{g}^+(x) \end{bmatrix}.$$
Since $x^*$ is regular, $E(x^*)$ has full column rank and the same is true for $E(x^k)$ if $k$ is sufficiently large. For such $k$, premultiplying (6) with $(E(x^k)^T E(x^k))^{-1}E(x^k)^T$, we obtain
$$ke(x^k) = -(E(x^k)^T E(x^k))^{-1}E(x^k)^T(\nabla f(x^k) + \alpha(x^k - x^*)).$$
By taking the limit as $k \to \infty$ and $x^k \to x^*$, we see that $\{ke(x^k)\}$ converges to the vector
$$\tau^* = -(E(x^*)^T E(x^*))^{-1}E(x^*)^T\nabla f(x^*).$$
By taking the limit as $k \to \infty$ in (6), we obtain
$$\nabla f(x^*) + E(x^*)\tau^* = 0.$$
Comparing against (4), we see that
$$\tau^* = \begin{bmatrix} \lambda^* \\ \mu^* \end{bmatrix};$$
in other words, $k\widetilde{g}^+(x^*) = \mu^*$. By the nonnegativity of $\widetilde{g}^+$, we obtain $\mu_j^* \ge 0$ for all $j$.

## 2 Additional Experiment Details

**Training.** All models were trained with mini-batch stochastic gradient descent (SGD) using a mini-batch size of 200. All weights were initialized from a zero-centered normal distribution with standard deviation 0.02. The dimension of the latent space is

- QM9: 128.
- ZINC: 256.
- Node-compatible: 128.

**Constraints.** Validity constraints for molecular graphs are ghost nodes, valence, and connectivity (i.e., (9) and (10)); whereas those for node-compatible graphs are ghost nodes and node compatibility (i.e., (9) and (11)). Regularization weights $\mu$ after tuning are

- QM9: ghost nodes/valence 1.0, connectivity 1.0.
- ZINC: ghost nodes/valence 0.05, connectivity 0.05.
- Node-compatible: ghost nodes 5.0, node compatibility 5.0.

**Metrics.** The experiment protocol generally follows that of Kusner et al. [1]. For "% Valid" and "% Novel", sample 1000 latent vectors from the prior and for each one, decode 500 times. For "% Recon.", set up a holdout set of 5000 graphs that do not participate training. For each graph in this set, encode 10 times and then decode 100 times. These steps done for the two baselines. For our proposed method, a slight difference is that in decoding, we perform maximum-likelihood decoding (which is equivalent to argmax). Since the decoding is deterministic, it is done only once for each latent vector.

## References

[1] Matt J. Kusner, Brooks Paige, and José Miguel Hernández-Lobato. Grammar variational autoencoder. In *Proceedings of the 34th International Conference on Machine Learning*, volume 70, pages 1945–1954, 2017.