[Reviews · NeurIPS 2018]

Reviewer 1



This paper proposes a generative model for molecule graphs based on a regularized VAE objective. The regularization is imposed on graphs generated from latent samples from the prior (and not those reconstructed from the approximate posterior). The model generates a larger % of valid graphs with respect to recent baselines. Pros: - The paper is very well written and clear, math is sound. - Methods seems to generate a larger % of valid graphs than the baselines. Cons: - Model outperforms baseline models on the metric it is practically trained to optimize (% of valid graphs when sampling from the generative model). Comments: - I would have loved to see some application of the model in downstream tasks, such as to better have an idea of the usefulness of the method , apart from generating more valid graphs. - Why Zinc dataset is so hard ? - I am not sure you need to call ‘z’ synthetic, the terminology ‘z’ sampled from the prior is widely understood. Typos: - In abstract, “For examples” -> For example

Reviewer 2



The main contribution of this paper is to formulate general structural constraints on the output of a generative model as regularization terms. The challenge is that the constraint is over the infinite set of all feasible outputs of the generator. To solve it, they marginalize over the outputs of the generator which is approximated by MC sampling. This is done in the framework of VAEs and as such marginalization is done over the latent value z using the reparametrization trick. The paper is well written. The notation is mostly clear. I have not checked the proof of the KKT constraint in the appendix but I have some comments about the formulation. The generalized notion of Lagrangian multipliers is very interesting. I'm wondering if the ramp function on the inequality constraints could be derived from the KKT conditions. Also, I don't think the KKT conditions have to be written using the squared constraints. Results are promising. There is one case where the proposed method does not perform well in reconstruction (Table 3-QM9 compared to GVAE). This needs more explanation.

Reviewer 3



This paper investigates a method for learning deep generative models of graphs, subject to constraints of the types that arise in many practical situations of interest. Rather than specify the model in such a way that it explicitly encodes valid graphs, instead a Lagrangian is formulated which penalizes generation of graphs which violate the constraints. In a VAE setting, the constraints themselves only apply to the generated graphs (x) not the latent variables (z). To ensure the constraints hold over the entire domain, the constraints are evaluated via Monte Carlo integration with values separately simulated from the prior p(z), rather than the values drawn from the approximating posterior q(z|x). The overall idea is fairly straightforward — Lagrange multipliers are the standard approach for converting constrained to unconstrained optimization problems. The most interesting aspect of this paper is in Theorem 1, which makes explicit that regularization of the VAE objective in this manner is appropriate. However, the exact form used is not justified particularly well in the main text. ** In particular: why in eq (6) would one consider the marginalization over the squared constraint, rather than the marginalization itself? Is this necessary for Theorem 1 to hold? The use of a ramp function in equation (7) seems a bit of a hack. ** If this were addressed more clearly, or better motivated, I would adjust my score upwards. There is also an overloading of notation which becomes a bit confusing, for the functions g_i(…, …). In equations (4) and (6), these are functions of x and z, while in e.g. equation (7) g_i is a function of theta and z. It would be good to separate the process which transforms a sampled z into a graph x (a process parameterized by theta) from the definitions of the constraints themselves, which typically seem to be only a function of the simulated x, not directly of z (the constraints are on graph properties, not on z). The experimental validation is fairly minimal, and does not compare against other graph generation methods, but rather only against VAEs which produce SMILES strings. I would think there should be some baselines for graph-based methods available for some of the proposed tasks. (For example, would it be possible to define exponential random graph models (ERGMs) which would include the constraints explicitly as likelihood terms, at the expense of easily simulating graphs?) As an example, the arXiv paper https://arxiv.org/abs/1805.09076 considers similar molecule generation tasks, but includes a couple explicitly graph-based baselines.